# DATA-FREE TRANSFORMER QUANTIZATION USING PARAMETER-SPACE SYMMETRY

## ABSTRACT

Transformer models are widely used in many learning tasks but incur large memory and compute costs, limiting their deployability. Post-Training Quantization (PTQ) is a promising solution but can lead to significant performance degradation. Many PTQ methods estimate weight and activation distributions with calibration data to account for outliers and maintain quantized performance. We propose a data-free approach to improve quantization by exploiting parameter space symmetries. We address outliers and high variability in weights by finding a transformation of the model weights that minimizes quantization error variance. Our approach is light-weight, data-free, and can be integrated as a pre-processing step within other PTQ methods. We evaluate our approach by testing quantized large language models on several benchmark tasks.

## 1 INTRODUCTION

Transformer models Vaswani et al. (2023) have found widespread success as generative models for language modeling and computer vision tasks. Transformers have become increasingly complex incurring large computational and memory storage costs far beyond other models, limiting their usability. The highest performing models have hundreds of billions of parameters Radford et al. (2019); Zhang et al. (2022) requiring immense training time and massive GPU memory. Even inference on pre-trained models can be prohibitively slow and exceed memory capacity of resource constrained systems. Effective model compression is essential for addressing these limitations.

Many model compression methods such as model-pruning Zhu et al. (2024) and low-bit quantization Chen et al. (2024); Ma et al. (2024) require re-training which is infeasible for models with billions of parameters. Post-training quantization (PTQ) which compresses models without re-training is a promising solution but can result in significant performance degradation. Many PTQ methods utilize calibration data and specialized heuristics to preserve model performance Bondarenko et al. (2021); Nagel et al. (2020). This requires access to high-quality calibration sets and can incur additional overhead for inference of the quantized model. Data-free methods for improving quantization performance have been proposed for MLPs and CNNs Meller et al. (2019); Nagel et al. (2019) but to our knowledge there are no similar methods for transformers.

In this paper, we develop a data-free method for improving post-training quantization of transformers by leveraging the symmetry of attention weights. Instead of designing a new quantization process, we provide a pre-quantization algorithm which finds equivalent weight configurations which are less sensitive to quantization. An equivalent weight configuration is a transformation of the weights which does not change the layer output. Our approach works by finding a linear transformation of the weights which minimizes the expected quantization error variance. This results in a new set of weights which when quantized results in lower quantization error during inference. There are several advantages to this strategy. First, we operate directly on the weights without any forward passes through the model. Second, our method is a pre-processing step which is compatible with any quantization algorithm allowing it to be stacked with existing techniques. This allows our method to be very lightweight, needing only enough memory for each layer's weights individually, while also being highly flexible and fast.

Our contributions include:

- A closed-form approximation of quantization error variance in attention.
- An optimization algorithm for finding optimal weight transformations.
- Empirical evaluation of our method showing its impact on linear quantization.

## 2 RELATED WORK

**Quantization of large language models (LLMs)**  Quantization reduces the numerical precision of neural network parameters to decrease model size and accelerate inference. This is essential for deploying LLMs efficiently across various hardware platforms. Common quantization techniques include quantization-aware training (QAT) and post-training quantization (PTQ) (Nagel et al., 2021; Zhu et al., 2024). QAT simulates quantization during training and adjusts model parameters to minimize quantization-induced error (Jacob et al., 2018; Esser et al., 2019). PTQ methods directly quantize pre-trained models. PTQ techniques include analytical methods that adjust weight distributions, such as range equalization and bias correction, enabling accurate quantization without access to training data (Nagel et al., 2019; Meller et al., 2019). Other PTQ approaches optimize quantization parameters on small calibration sets (Nagel et al., 2020; Hubara et al., 2021; Li et al., 2021). Recently, PTQ has become prominent for quantizing transformers and large language models (Frantar et al., 2022; Yao et al., 2022; Xiao et al., 2023; Dettmers et al., 2022). Our work follows the post-training quantization paradigm, aiming to further reduce quantization-induced accuracy loss through optimized parameter symmetry transformations.

**Using symmetry in quantization**  Neural networks often have parameter space symmetries, meaning certain transformations of their parameters leave the network's loss unchanged. Examples include the scaling symmetry in networks with ReLU or linear activations (Badrinarayanan et al., 2015), and permutation symmetry among neurons within a hidden layer (Hecht-Nielsen, 1990). Several works have explicitly used such weight transformations to reduce quantization error. A common strategy is to exploit scale invariances to adjust the range of weights or activations before quantization. For example, Nagel et al. (2019) and Meller et al. (2019) propose equalizing weight ranges across layers in ReLU-based networks using the scaling symmetry. Xiao et al. (2023) improve speed and reduce memory during inference for linear operations, defined as computing the product of activations (output from previous computations) and weights, by applying a loss-invariant scaling on both parts before quantization. While this transformation is defined jointly on parameters and activations, it can be expressed as a parameter symmetry when the activation is the output of a linear operation. Similarly, Kim et al. (2024) scales activation and weights in CNN-transformer hybrid architectures to align parameter distributions with hardware-friendly quantization constraints, thereby improving inference efficiency. Our approach extends these ideas by considering the full general linear group, optimizing over a broader class of symmetry transformations to achieve superior quantization accuracy.

**Optimization in transformer model level sets**  Recent works have also explored optimization over the loss level sets in transformers for applications other than quantization. This optimization is often done on symmetry group orbits, leveraging the general linear group symmetry in self-attention layers. For example, Zhang et al. (2025) improves model fusion by minimizing the distance between two self-attentions without affecting their loss. Their method first finds an optimal rotation of key and query matrices, followed by an optimal scaling. Similarly, Wu et al. (2025) accelerates the training of transformers by searching in the loss level set for points better suited for optimization. We also optimize over the symmetry group orbits of transformer models, but with the specific goal of finding transformations that minimize accuracy loss in quantization.

## 3 BACKGROUND

### 3.1 TRANSFORMER ATTENTION

A standard transformer layer consists of two main modules: a multi-head attention(MHA) module and a multi-layer perceptron(MLP). In this work we focus on improving the quantization of the

attention module. The attention module has four weight matrices $W_q, W_k, W_V, W_O$. For a given transformer layer with input $x \in \mathbb{R}^{n \times d}$ the attention scores are computed as:

$$A = xW_qW_k^Tx^T \tag{1}$$

A softmax is applied after to normalize the scores and the final layer output is computed:

$$\text{MHA}(x) = \text{softmax}\left(\frac{A}{\sqrt{d}}\right)xW_VW_O \tag{2}$$

We focus on quantizing $W_q, W_k$ although we believe our results may be generalized to include $W_v$ and $W_O$.

## 3.2 QUANTIZATION

At a high-level, quantization works by mapping full-precision floating point values into a smaller set of low-bit numbers (e.g. 8-bit, 4-bit integers). The low-bit numbers are used during computation and then the resulting output is reconstructed by de-quantization which uses the inverse map to recover the approximate floating point value.

**Uniform Quantization**  A common mapping used in quantization is uniform quantization. Uniform quantization splits the range $R$ of a tensor $Y$ uniformly onto a set of $b$-bit signed integers. The range $R$ can be defined for symmetric quantization or asymmetric quantization. For symmetric quantization $R$ is defined as the maximum absolute value $\max(|Y|)$ while asymmetric quantization sets $R$ as $\max(Y) - \min(Y)$. Quantization is defined as:

$$\text{Quant}(Y) = \text{Clamp}\left(\text{Round}\left(\frac{Y2^{b-1}}{R}\right), -2^{b-1}, 2^{b-1} - 1\right) \tag{3}$$

Quantization error is computed between the original tensor $Y$ and the de-quantized reconstruction $\hat{Y}$. $\hat{Y}$ is obtained by the inverse map $\text{DeQuant}() = \text{Quant}^{-1}()$. Since quantization is surjective, there can be errors in the reconstruction. We write this element-wise quantization error as $\Delta Y = \hat{Y} - Y$. The full tensor quantization error is defined as the L2 norm of the per-element error $||\Delta Y||_2^2 = |\Delta Y|^2$.

Uniform quantization depends heavily on the range $R$. When $R$ is lower than the actual range of values in the tensor, extremal values are clamped leading to large and inconsistent quantization error. Conversely a large range results in a lower resolution mapping leading to higher uncertainty in the reconstruction for all values. This means quantization error is driven primarily by the extremal values of $Y$, so outlier values can dramatically impact quantization. Under uniform quantization, the quantization error is approximately distributed uniformly Marco & Neuhoff (2005); Lin et al. (2016):

$$\Delta Y \sim \text{Uniform}\left(\frac{-R}{2^b}, \frac{R}{2^b}\right) \tag{4}$$

**Neural Network Quantization**  Neural network quantization consists of two parts. Weight quantization is quantization applied to the model weights. The simplest form of weight quantization is per-tensor where $R$ is computed for each individual weight tensor which are quantized independently. Activation quantization is quantization of the layer outputs and is similarly applied per-tensor.

Weight quantization is straightforward since the weights are fixed and $R$ is easily computed directly from the weights. By contrast, activation quantization requires calibration by estimating $R$ over a calibration dataset of inputs. The layer outputs depend on layer inputs so the range of values are not known a priori. If the calibration set does not capture the true range of activations seen during inference the estimated range $R$ will be too small resulting in high quantization error. This sensitivity to calibration is especially impactful in settings where the training data is not accessible or when there is a limited amount of calibration data.

## 4 DATA-FREE ESTIMATION OF QUANTIZATION NOISE

To improve quantization without data, we compute an analytic expression for the quantization noise, which gives a data-free objective for minimizing the error under quantization. Intuitively quantization

noise is an estimate of the variance of quantization error seen during inference. We use this as a data-free objective for reducing the calibration sensitivity of activation quantization.

This approach is inspired by Meller et al. (2019). Meller et. al. develop a greedy algorithm for re-scaling weights of CNNs to minimize quantization noise of activations. Our method uses a similar objective but differs in two ways. Our approach is data-free without requiring estimations for activation ranges and considers orthogonal transforms of the weights instead of rescaling.

**Quantization Error** Let $Y = W_q W_k^T$ which when quantized and reconstructed gives $\hat{Y} = \hat{W}_q \hat{W}_k^T$. Rewriting this in terms of the quantization error we get an expression for $\Delta Y$:

$$
\begin{aligned}
Y + \Delta Y =& (W_q + \Delta W_q)(W_k + \Delta W_k)^T \\
\Delta Y =& W_q \Delta W_k^T + \Delta W_q W_k^T + \Delta W_q \Delta W_k^T
\end{aligned}
\tag{5}
$$

The element-wise quantization errors $\Delta W_q, \Delta W_k$ are both random tensors approximately distributed as:

$$
\Delta W_q \sim \text{Uniform}\left(\frac{-R_q}{2^{b+1}}, \frac{R_q}{2^{b+1}}\right)
\tag{6}
$$

$$
\Delta W_k \sim \text{Uniform}\left(\frac{-R_k}{2^{b+1}}, \frac{R_k}{2^{b+1}}\right)
\tag{7}
$$

where $R_q, R_k$ are the ranges of $W_q, W_k$ respectively and $b$ is the quantization bit-width. This means $\Delta Y$ is also a random tensor which depends on $\Delta W_q, \Delta W_k$.

**Quantization Noise Objective** We define the quantization noise as the average element-wise variance $\text{mean}(\mathbf{E}(|\Delta Y|^2))$ which is the expected magnitude of the full tensor quantization error. Intuitively higher quantization noise corresponds to higher uncertainty in the de-quantized reconstruction $\hat{Y}$ which is driven by outliers which pose significant challenges to effective quantization. This makes minimizing quantization noise a promising data-free objective that can lead to fewer outliers and better quantization.

We now show how to compute the quantization noise, for a full proof see Appendix A. In what follows $\odot$ is element-wise multiplication. Expanding and simplifying $\mathbf{E}(|\Delta Y|^2)$ yields a sum over over 6 term matrices. Equations for each of these terms is included in Appendix 1.

**Lemma 4.1.**

$$
\begin{aligned}
\mathbf{E}(|\Delta Y|^2) = \mathbf{E}[& |W_q \Delta W_k^T|^2 + |\Delta W_q W_k^T|^2 \\
& + |\Delta W_q \Delta W_k^T|^2 \\
& + |W_q \Delta W_k^T| \odot |\Delta W_q W_k^T| \\
& + |W_q \Delta W_k^T| \odot |\Delta W_q \Delta W_k^T| \\
& + |\Delta W_q W_k^T| \odot |\Delta W_q \Delta W_k^T|]
\end{aligned}
\tag{8}
$$

Since we only need the element-wise mean of this matrix expression, these terms can be further reduced giving the following proposition.

**Proposition 4.2.** *Let $W_q, W_k \in \mathbb{R}^{n \times m}$ with elements denoted $q_{ij}, k_{ij}$. Let $\Delta W_q, \Delta W_k$ be their quantization error matrices respectively. If $\Delta W_q \sim \text{Uniform}(-r_q, r_q)$ and $\Delta W_k \sim \text{Uniform}(-r_k, r_k)$ then the mean of the elements in the matrix expression in Equation 8 is:*

$$
\frac{r_k^2}{n}\left(\frac{\sum_{i,j} q_{ij}^2}{12} + \frac{\sum_{i,j,t} q_{ij} q_{it}}{4}\right) + \frac{r_q^2}{n}\left(\frac{\sum_{i,j} k_{ij}^2}{12} + \frac{\sum_{i,j,t} k_{ij} k_{it}}{4}\right)
$$

$$
+ \frac{m r_q^2 r_k^2}{16}\left(m + \frac{7}{9}\right) + \frac{r_q r_k}{2n^2}\left(\sum_{i,j} q_{ij}\right)\left(\sum_{i,j} k_{ij}\right)
$$

$$
+ \frac{r_q r_k^2 (3m+1)}{12n}\sum_{i,j} q_{ij} + \frac{r_q^2 r_k (3m+1)}{12n}\sum_{i,j} k_{ij}
$$

*where the summands correspond to those in Equation 8.*

## 5 METHOD

In this section we introduce our algorithm for minimizing quantization noise. Our approach finds a transformation of $W_q, W_k$ which minimizes the quantization noise without changing the layer function through gradient descent using the equation in 4.2 as an objective. At a high-level, we parameterize an orthogonal matrix $g$ for each layer which can be optimized independently or batched to be feasible on wide variety of hardware. Our approach only modifies the weights of the attention layers in a transformer without impacting model behavior or specialized quantization heuristics. It is therefore compatible as a pre-processing step with any downstream quantization algorithm that uses uniform quantization.

From Equation 1, the attention scores are computed as $A = xW_qW_k^Tx^T$ where $W_q, W_k \in \mathbb{R}^{n \times m}$. An invertible matrix $g \in \mathbb{R}^{m \times m}$ and its inverse can be inserted between $W_q$ and $W_k$ giving an equal attention score:

$$A = xW_qgg^{-1}W_k^Tx^T \tag{9}$$

Replacing the original weights with $W_q' = W_qg$, and $W_k' = W_k(g^{-1})^T$ gives a new set of weights without changing the layer functionally.

Our goal is to find such a transformation $g$ which minimizes the quantization noise for the new weights. Instead of searching over the group $\text{GL}(m)$, all invertible $m \times m$ matrices, we restrict $g$ to be orthogonal. The group $O(m)$ is compact, which assures the existence of a global minimum, making the optimization problem well posed. This approach compliments weight rescaling which is a common technique for improving quantization Nagel et al. (2019); Meller et al. (2019); Xiao et al. (2023) by exploring a different subspace of parameter symmetries. Due to orthogonality the new weights are $W_q' = W_qg$ and $W_k' = W_kg$. Our objective more concretely is to solve the following minimization:

$$g = \text{argmin}_{g \in O(m)}\text{mean}(\mathbf{E}(\Delta Y'^2)) \tag{10}$$

This is solvable by gradient descent using the expression from Proposition 4.2 as a loss function. We parameterize $g$ by instantiating a square random matrix $M$ and setting $g$ as the orthogonal component of the QR decomposition $\text{QR}(M)$. Since the QR decomposition is differentiable, this makes for a suitable parameterization. We perform this procedure for each layer of the transformer model and for each head in multi-headed attention layers which can be batched to improve efficiency.

Our method ensures that the product of queries and keys is maintained resulting in identical attention scores but the individual weight matrices $W_q, W_k$ are changed to $W_q', W_k'$. The transformed weights will have different ranges and therefore have different quantization error from weight quantization. Similarly the query $Q = xW_q$ and key $K = xW_k$ values will also be transformed to $Q', K'$. The transformed values will in general have different quantization errors from activation quantization. Quantization of the intermediate $Q, K$ tensors is not always performed Yao et al. (2022) in which case the activation quantization will not be impacted by our transformation beyond the impact to activations due to weight quantization.

**Analyzing Impact of Learned Transformations** We begin by comparing the product $QK^T$ for the original weights and the transformed weights. Figure 1a shows that our method indeed produces identical attention scores at full-precision. This ensures that the output for every layer in the model is identical and so the overall model behavior is maintained for any input.

Weight quantization however is changed after transformation since the original weight matrix ranges $R_q, R_k$ are different from the transformed ranges $R_q', R_k'$. Since the quantization error of weight quantization depends on these ranges, the transformed weights will have different quantization performance despite the functional equivalence of the full-precision model. We compare the ranges before and after transformation in figure 1b and find that the learned transformation reduces the weight ranges for both queries and keys which results in lower quantization error on average.

The query activations $Q$ and key activations $K$ are also changed since our transformation only maintains functional equivalence of the product $QK^T$. This means our transformation will also impact activation quantization error. We compare the activation ranges over a set of 100 inputs in figure 2. We find that our transformed weights generally reduced the ranges for query activations by 15.3% and for key activations by 4.1% averaged over all layers. There are some layers where the

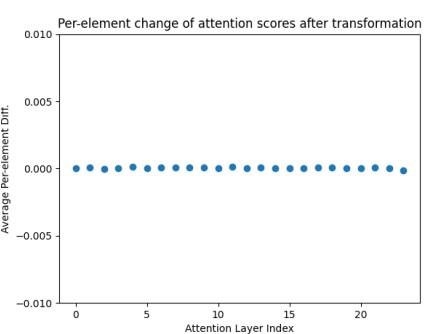

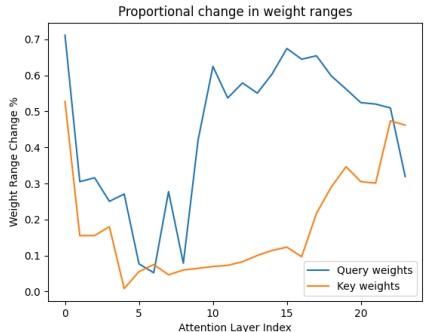

(a) Average per-element difference of products $W_q W_k^T$ and $W_q' W_k'^T$ for OPT$_{1.3b}$. The average per-element change over all layers is $3.8 \times 10^{-5}$ showing that the transformation $g$ does not change the attention scores.

(b) Relative change in weight ranges for OPT$_{1.3b}$. Ranges are reduced on average by $44.4\%$ for $W_q$ and $18.2\%$ for $W_k$. The reduction in weight ranges after transformation results in lower weight quantization error.

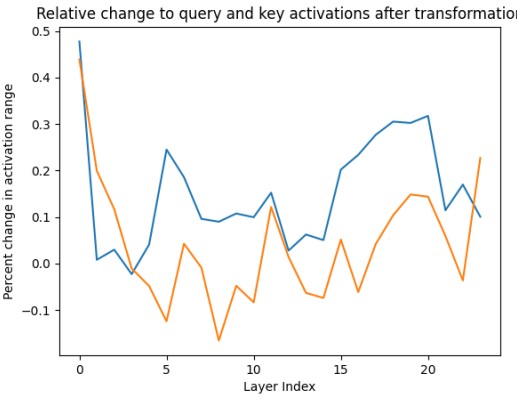

Figure 2: Percentage change in activation ranges for queries $Q$ and keys $K$ from transformed weights over 100 input sentences. Ranges are reduced on average by $15.3\%$ for $Q$ and $4.1\%$ for $K$. This shows that the learned transformation reduces the activation ranges despite being optimized data-free.

range is slightly increased however. This shows that the data-free quantization noise objective is a promising metric for outlier reduction even for data dependent activations.

## 6 EXPERIMENTAL EVALUATION

**Setups** As a proof of concept, we tested our approach by validating the relative performance impact of quantization with and without our transformation. We compare text classification accuracy for Bert$_{base}$ Devlin et al. (2018) fine-tuned for two benchmark GLUE tasks, SST-2 and MNLI. In addition we compare text generation performance of OPT$_{350m}$ and OPT$_{1.3b}$ by measuring perplexity over the WikiText-2 dataset. For text generation we evaluate both the fine-tuned performance as well as zero-shot performance.

**Implementation** We implement both symmetric per-tensor weight quantization and symmetric per-tensor activation quantization. In our experiments we test both 4-bit and 8-bit weight quantization. We also add 8-bit activation quantization for the text generation experiments. These are denoted W$x$A$y$ where $x$ is the bits for weight quantization and $y$ is the bits for activation quantization. The full-precision models use 16-bit floating points so W16 and A16 refer to no quantization for weights or activations respectively.

**Text Classification Results** We used Bert$_{base}$ Devlin et al. (2018) fine-tuned for two benchmark GLUE tasks, SST-2 and MNLI. The model weights were quantized to 8-bit and 4-bit integers without

| Model | SST-2 - Acc. | MNLI - Acc. |
|---|---|---|
| Full-Prec. | 92.2% | 84.1% |
| Baseline W8A16 | 91.9% | 84.1% |
| **Transformed W8A16** | 91.9% | 84.1% |
| Stand. 4-bit | 91.7% | 84.0% |
| **Transformed W4A16** | 91.9% | 84.1% |

Table 1: MNLI and SST-2 quantization performance results. Baseline W8A16 and W4A16 were quantized without weight transformation. Transformed W8A16 and W4A16 applied weight transformation before quantization.

any activation quantization. The transformation optimization was run for 5,000 iterations for both tasks. The results are summarized in Table 1. W8A16 and W4A16 weight quantization did not degrade performance nearly at all for either task suggesting the prediction accuracy is not very sensitive to weight quantization. We see a marginal improvement in quantization performance using the transformed weights but the generally low impact of weight quantization suggests a more sensitive metric and task are needed to evaluate our approach.

**Text Generation Results**  We compared quantization performance of $OPT_{350m}$ and $OPT_{1.3b}$ over the WikiText-2 dataset. Perplexity is much more sensitive to quantization which is confirmed in the results of other quantization research Xiao et al. (2023). We use a calibration set of 10 randomly selected sentences from the training set for activation quantization.

| | Fine-tuned Perplexity | | | Zero-Shot Perplexity | | |
|---|---|---|---|---|---|---|
| Model | Full-Prec. | W8A8 | W4A8 | Full-Prec. | W8A8 | W4A8 |
| OPT-350m | 17.9 | 355.3 | 3853.4 | 23.6 | 472.8 | 5378.1 |
| Transf. OPT-350m | 17.9 | 254.0 | 3235.7 | 23.6 | 446.4 | 5217.9 |
| OPT-1.3b | 12.3 | 4284.8 | 9097.0 | 15.3 | 7804.9 | 10392.9 |
| Transf. OPT-1.3b | 12.3 | 4193.2 | 8844.8 | 15.3 | 7116.6 | 10242.5 |

Table 2: Fine-tuned and zero-shot text generation perplexity results evaluated over WikiText-2. The transformed weights improve the performance for quantized models over the standard uniform symmetric quantization baseline. The performance is identical for full-precision models due to the functional equivalency of the transformed weights.

Table 2 shows the fine-tuned performance of the models. We find that the transformed model substantially improves quantization performance for both models. This improvement is relatively stronger for W8A8 quantization than W4A8 quantization where the relative impact is reduced.

Table 2 shows the zero-shot performance of the models which mirrors the previous fine-tuned performance. The baseline performance is lower than the fine-tuned models with much lower quantized performance. This suggests the zero-shot performance is more sensitive to quantization error. The transformation again improves quantization performance but with a lower impact than the fine-tuned models.

These results show that our method can consistently reduce quantization error. The lower relative impact for high quantization error may be caused by the propagation of large quantization errors in early layers which reduce the impact of the weight transformations for later layers. The weight transformations for later layers are computed on the full-precision weights which does not account for the accumulation of quantization errors from earlier layers resulting in a larger difference between the inputs to full-precision and quantized layers.

## 7    CONCLUSION AND FUTURE WORK

In this paper we explored using parameter symmetries to improve quantization. We derived an estimate for quantization noise in query and key attention. Our approach for minimizing quantization noise is a highly efficient pre-processing step which is compatible with other downstream quantization approaches and consistently reduces weight ranges and on average reduces query and key activation

ranges. In quantization sensitive tasks such as text generation, the learned weight transformations improve quantization performance however the impact is limited for high quantization error where the quantized values differ significantly from the full-precision values we optimize over.

In the future we plan to generalize our quantization noise estimation to per-group and per-channel quantization which may provide a more fine-grained estimate and more effective weight transformations. Removing the orthogonal transformation restriction is another future direction which contains both our current orthogonal parameter symmetry as well as the common weight rescaling symmetry. Exploring the parameter symmetries of the value and output weight matrices may also lead to more consistent and larger improvements to quantization.

Finally, the quantization noise estimate may be a useful metric for quantization aware training (QAT). Our approach for minimizing quantization noise can be used to transform weights during training similarly to teleportation Wu et al. (2025).

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

# A  QUANTIZATION NOISE ESTIMATION PROOF

In this section we provide a proof of the equations found in proposition 4.2. In the following proofs, $Q^i, K^i$ denote the $m$-dimensional $i$-th row vectors of $W_q, W_k$ and $\delta Q^i, \delta K^i$ are the rows of $\Delta W_q, \Delta W_k$.

**1st Term:** $|W_q \Delta W_k^T|^2$   We begin by first considering the matrix $\mathbf{E}[|W_q \Delta W_k^T|^2]$. The value $\mathbf{E}[|W_q \Delta W_k^T|_{ij}^2]$ at index $i, j$ is computed as follows:

$$\mathbf{E}[|W_q \Delta W_k^T|_{ij}^2] = \mathbf{E}[|Q^i (\delta K^j)^T|^2]$$

$$= \mathbf{E}[|\sum_u Q_u^i \delta K_u^j| \cdot |\sum_v Q_v^i \delta K_v^j|]$$

Expanding this product, the expectation can be distributed through the sum. In this expanded product there are 2 cases, when $u = v$ and $u \neq v$.

When $u = v$, this gives $\mathbf{E}[|Q_u^i \delta K_u^j|^2] = (Q_u^i)^2 \frac{r_k^2}{3}$ since $|\delta K_u^j| \sim \text{Uniform}(0, r_k)$. Since $u, v$ go from 1 to m, this will give us $\frac{r_k^2}{3} \sum_{u=1}^m (Q_u^i)^2$ in the sum.

When $u \neq v$, the value is $\mathbf{E}[|Q_u^i \delta K_u^j|] \mathbf{E}[|Q_v^i \delta K_v^j|]$ since $\delta K_u^j$ and $\delta K_v^j$ are independent random values so their expectations are multiplied. This gives $\frac{r_k^2}{4} \sum_{u \neq v} Q_u^i Q_v^i$.

Putting both cases together we get the final value for index $i, j$ of

$$\mathbf{E}[|W_q \Delta W_k^T|_{ij}^2] = \frac{r_k^2}{3} \sum_{u=1}^n (Q_u^i)^2 + \frac{r_k^2}{4} \sum_{u \neq v} Q_u^i Q_v^i$$

$$= \frac{r_k^2}{3} \sum_{u=1}^n (Q_u^i)^2 + \frac{r_k^2}{4} \sum_{u,v} Q_u^i Q_v^i - \frac{r_k^2}{4} \sum_{u=1}^n (Q_u^i)^2$$

$$= \frac{r_k^2}{12} \sum_{u=1}^n (Q_u^i)^2 + r_k^2 \sum_u Q_u^i \sum_v Q_v^i$$

Note that this final value does not depend on $j$ meaning all of the values in row $i$ will have this value giving us a total of $n$ copies.

We now take the average over the $n^2$ values in $\mathbf{E}[|W_q \Delta W_k^T|_{ij}^2]$ which gives us the desired form:

$$\text{mean}(\mathbf{E}[|W_q \Delta W_k^T|^2]) = \frac{r_k^2}{n} \left( \frac{\sum_{i,j} q_{ij}^2}{12} + \frac{\sum_i (\sum_{j,t} q_{ij} q_{it})}{4} \right)$$

**2nd Term:** $|\Delta W_q W_k^T|^2$   Following the same reasoning as the previous term, the value $\mathbf{E}[(\Delta W_q W_k^T)_{ij}^2]$ is:

$$\mathbf{E}[|\Delta W_q W_k^T|_{ij}^2] = \mathbf{E}[|\delta Q^i (K^j)^T|^2]$$

$$= \mathbf{E}[|\sum_u \delta Q_u^i \delta K_u^j| \cdot |\sum_v \delta Q_v^i K_v^j|]$$

The exact same simplifications as before occur but since $\delta Q^i$ is the random vector, we instead will get a formula which does not depend on $i$:

$$\mathbf{E}[|W_q \Delta W_k^T|_{ij}^2] = \frac{r_q^2}{12} \sum_{u=1}^n (K_u^j)^2 + \frac{r_q^2}{4} \sum_u K_u^j \sum_v K_v^j$$

Taking the average over the $n^2$ values gives the final form:

$$\text{mean}(\mathbf{E}[|\Delta W_q W_k^T|^2]) = \frac{r_q^2}{n} \left( \frac{\sum_{i,j} k_{ij}^2}{12} + \frac{\sum_i (\sum_{j,t} k_{ij} q_{it})}{4} \right)$$

**3rd Term:** $|\Delta W_q \Delta W_k^T|^2$   This case is much easier since the values of $\Delta W_q, \Delta W_k^T$ are i.i.d. and so every value of the matrix $\mathbf{E}[|\Delta W_q \Delta W_k^T|^2]$ are equal. A single value of this matrix is computed:

$$\mathbf{E}[|\Delta W_q \Delta W_k^T|_{ij}^2] = |\sum_u \delta Q_u^i \delta K_u^j| \cdot |\sum_v \delta Q_v^i \delta K_v^j|$$

In the first case $u = v$, the result is $\mathbf{E}[|\delta Q_u^i \delta K_u^j|^2] = \frac{r_q^2 r_k^2}{9}$. This will happen $m$ times since $u, v$ go from 1 to $m$.

The second case $u \neq v$ gives $\mathbf{E}[|\delta Q_u^i \delta K_u^j| \cdot |\delta Q_v^i \delta K_v^j|] = \frac{r_q^2 r_k^2}{16}$. This happens for when $u \neq v$ so we will have this $m(m-1)$ times in the sum.

Putting these two together we get a simplified per element value of:

$$\mathbf{E}[|\Delta W_q \Delta W_k^T|_{ij}^2] = m\frac{r_q^2 r_k^2}{9} + (m^2 - m)\frac{r_q^2 r_k^2}{16}$$

$$= \frac{m r_q^2 r_k^2}{16}(m + \frac{7}{9})$$

The average value is exactly equal to the per element value since every element is equivalent under expectation.

**4th Term:** $|W_q \Delta W_k^T| \odot |\Delta W_q W_k^T|$   Once again begin with the $i, j$ entry of the matrix:

$$\mathbf{E}[|W_q \Delta W_k^T| \odot |\Delta W_q W_k^T|_{ij}] = \mathbf{E}[|Q^i (\delta K^j)^T| \cdot |\delta Q^i (K^j)^T|]$$

$$= \mathbf{E}[|\sum_u Q_u^i \delta K_u^j| \cdot |\sum_v \delta Q_v^i K_v^j|]$$

$$= (m\frac{r_k}{2}\sum_u Q_u^i)(m\frac{r_q}{2}\sum_v K_v^j)$$

Averaging over all $i, j$ elements gives the final form:

$$\text{mean}(\mathbf{E}[|W_q \Delta W_k^T| \odot |\Delta W_q W_k^T|]) = \frac{r_q r_k}{2n^2}(\sum_{i,j} q_{ij})(\sum_{i,j} k_{ij})$$

**5th Term:** $|W_q \Delta W_k \odot \Delta W_q \Delta W_k|$

$$\mathbf{E}[|W_q \Delta W_k^T| \odot |\Delta W_q \Delta W_k^T|_{ij}] = \mathbf{E}[|Q^i (\delta K^j)^T| \cdot |\delta Q^i (\delta K^j)^T|]$$

$$= \mathbf{E}[|\sum_u Q_u^i \delta K_u^j| \cdot |\sum_v \delta Q_v^i \delta K_v^j|]$$

Once again there are 2 cases when $u = v$ and when $u \neq v$.   In the first case $\mathbf{E}[(Q_u^i \delta K_u^j)(\delta Q_u^i \delta K_u^j)] = \frac{r_q}{2}\frac{r_k^2}{3}Q_u^i$. In the second case the random values are all independent so the result is: $\mathbf{E}[(Q_u^i \delta K_u^j)(\delta Q_u^i \delta K_v^j)] = \frac{r_q}{2}\frac{r_k^2}{4}Q_u^i$.

Adding this up and simplifying gives the value for element $i, j$:

$$\mathbf{E}[|W_q \Delta W_k^T| \odot |\Delta W_q \Delta W_k^T|_{ij}] = m\frac{r_q r_k^2}{6}\sum_u Q_u^i + (m^2 - m)\frac{r_q r_k^2}{8}\sum_u Q_u^i$$

Averaging over all $i, j$ elements gives:

$$\text{mean}(\mathbf{E}[|W_q \Delta W_k^T| \odot |\Delta W_q \Delta W_k^T|]) = \frac{r_q r_k^2(3m+1)}{12n}\sum_{i,j} q_{ij}$$

**6th Term:** $|\Delta W_q W_k^T| \odot |\Delta W_q \Delta W_k^T|$   This term follows the same reasoning as above. Starting with entry $i, j$:

$$\mathbf{E}[|\Delta W_q W_k^T| \odot |\Delta W_q \Delta W_k^T|_{ij}] = \mathbf{E}[|\delta Q^i (K^j)^T| \cdot |\delta Q^i (\delta K^j)^T|]$$

$$= \mathbf{E}[|\sum_u \delta Q_u^i K_u^j| \cdot |\sum_v \delta Q_v^i \delta K_v^j|]$$

In the case where $u = v$ we get $\mathbf{E}[(\delta Q_u^i K_u^j)(\delta Q_u^i \delta K_u^j)] = \frac{r_q^2}{3} \frac{r_k}{2} K_u^j$. Similarly for $u \neq v$ gives $\mathbf{E}[(\delta Q_u^i K_u^j)(\delta Q_u^i \delta K_v^j)] = \frac{r_q^2}{4} \frac{r_k}{2} K_u^j$.

Adding both cases up and simplifying gives:

$$\mathbf{E}[|\Delta W_q W_k^T| \odot |\Delta W_q \Delta W_k^T|_{ij}] = m \frac{r_q^2 r_k}{6} \sum_u K_u^j + (m^2 - m) \frac{r_q^2 r_k}{8} \sum_u K_u^j$$

Averaging over all $i, j$ elements gives the final equation:

$$\text{mean}(\mathbf{E}[|\Delta W_q W_k^T| \odot |\Delta W_q \Delta W_k^T|]) = \frac{r_q^2 r_k (3m + 1)}{12n} \sum_{i,j} k_{ij}$$

