# OpenReview forum: "Data-Free Transformer Quantization Using Parameter-Space Symmetry"
_ICLR.cc/2026/Conference — ICLR 2026 Conference Withdrawn Submission_

### Official Review · Reviewer_HgyE · 2025-10-19

**Soundness:** 2
**Presentation:** 1
**Contribution:** 2
**Rating:** 4
**Confidence:** 4

**Summary:**

This paper proposes a data-free optimization method for post-training quantization (PTQ) of Transformer models by exploiting parameter-space symmetries inherent in the attention mechanism to reduce quantization error. The authors observe that, in standard Transformers, the query $W_q$ and key $W_k$ weight matrices can be reparameterized via orthogonal transformations without altering the model output, provided the product $W_q{W_k}^⊤$ remains invariant. Building upon this observation, the paper derives a closed-form approximation of the quantization error variance and constructs a calibration-free quantization noise objective function. By optimizing this objective over the orthogonal group, the proposed method effectively compresses the dynamic ranges of both weights and activations, thereby mitigating performance degradation caused by outlier values during quantization. Experiments demonstrate that the proposed preprocessing strategy can be seamlessly integrated into existing post-training quantization (PTQ) pipelines and consistently improves performance under both 4-bit and 8-bit quantization across text classification and generation tasks on models such as BERT and OPT. The gains are particularly pronounced on perplexity—a metric highly sensitive to quantization artifacts. Owing to its lightweight nature, computational efficiency, and decoupling from specific model architectures, the method offers a novel and practical pathway toward data-free, efficient deployment of large language models.

**Strengths:**

1. By leveraging an analytically derived expression for quantization noise that depends solely on model weights, without requiring calibration or training data, proposed method enables optimization while circumventing the high costs and privacy concerns associated with data acquisition.

2. The proposed method applies only orthogonal transformations to the attention weights, requiring no forward passes through the model and thus incurring minimal memory overhead. It can be seamlessly integrated as a preprocessing step with any uniform quantization–based post-training quantization (PTQ) method, without interfering with or altering the original quantization logic.

3. The method demonstrates pronounced effectiveness on quantization-sensitive tasks such as text generation, significantly reducing model perplexity—for instance, lowering the W8A8 perplexity of OPT-350m from 355.3 to 254.0

**Weaknesses:**

1. The orthogonal transformation technique proposed in the paper has precedent in prior works such as QuIP and QuaRot. However, the current work applies this transformation narrowly—specifically to optimize only the query and key weight matrices in the attention module. It formulates a variance-based quantization error objective derived from the matrix multiplication output. While technically sound, the scope of innovation is limited, as the method offers no additional contributions or extensions to other components of the Transformer architecture beyond this specific use case.

2. Although the paper claims to reduce quantization error by minimizing a "quantization noise" objective(element-wise variance mean), empirical results reveal limited practical efficacy. In text classification tasks, the proposed optimization yields negligible accuracy gains under both 4-bit and 8-bit quantization—for instance, MNLI accuracy under W4A16 improves only marginally from 84.0% to 84.1%. Similarly, in text generation, the method’s effectiveness substantially diminishes in high-quantization-error regimes (e.g., W4A8), where performance gains are markedly attenuated. These observations suggest that the derived quantization noise objective fails to faithfully capture or correlate with the actual quantization-induced loss, indicating a misalignment between the theoretical formulation and real-world quantization behavior.

3. The primary experiments are conducted predominantly on OPT-family models and do not include evaluations on more widely adopted architectures such as LLaMA or Qwen. Moreover, the tested models are generally of smaller scale, limiting the generalizability of the findings to larger, state-of-the-art language models. The empirical comparison also omits strong contemporary baselines, including QuaRot, GPTQ, and AWQ, making it difficult to assess the relative effectiveness of the proposed method. Furthermore, the evaluation relies solely on perplexity (PPL) as the performance metric, leaving open the question of whether similar improvements would hold across other downstream tasks or application scenarios.

**Questions:**

1. To rigorously validate the generalizability and practical utility of the proposed method, it is essential to demonstrate its effectiveness on larger-scale, widely adopted architectures—specifically, LLaMA and Qwen series models with at least 7 billion parameters. Furthermore, comprehensive ablation studies should be conducted against strong contemporary quantization baselines such as GPTQ, AWQ, and QuaRot. Evaluation should extend beyond perplexity to include generative downstream tasks that are sensitive to quantization artifacts, such as GSM8K for mathematical reasoning and HumanEval for code generation, to assess whether the method consistently preserves functional capabilities under aggressive quantization.

2. How can the effectiveness of the proposed quantization noise, used as the optimization objective in the paper, be rigorously justified? Specifically, can the authors provide a theoretical or empirical analysis that characterizes its geometric or functional differences from conventional objectives such as mean squared error (MSE) or KL divergence? Such a comparison would clarify whether the proposed objective leads to a more favorable optimization landscape, e.g., smoother curvature, fewer spurious local minima, or better alignment with actual quantization-induced loss, thereby enabling faster convergence or yielding solutions that generalize better under quantization.

3. There are two matrix multiplications (matmuls) in the attention component of the LLM architecture. Regarding the weight transformation proposed in the paper for $W_q$ and $W_k$, can it be extended to the second matmul (i.e., incorporating $W_v$) or to the element-wise multiplication in the feed-forward network (FFN) component? If feasible, could the formula derivation be provided?

---

### Official Review · Reviewer_Gy5X · 2025-10-30

**Soundness:** 3
**Presentation:** 2
**Contribution:** 2
**Rating:** 2
**Confidence:** 4

**Summary:**

The authors propose a data-free preprocessing step for PTQ methods by minimizing quantization error variance using a learned transformation. Experiments are conducted on BERT and OPT family of models.

**Strengths:**

- Using the variance of quantization noise as an objective is an interesting idea
- The authors provide theoretical derivation for the minimization objective.

**Weaknesses:**

- Experiments are only conducted on small scale models like BERT and OPT which have a very different distribution of weights compared to the newer generations of language models.
- The proposed methodology isn't novel and needs more experimentation to demonstrate it's efficacy
- The authors say this can be used in conjunction with other PTQ methods but do not demonstrate it's impact on them. Can we see some results on different types of PTQ methods.
- The authors mostly measure perplexity which has weak correlation with downstream performance. It would be nice to see the impact on real complex benchmarks which reflect the degradation better.
- The degradation due to quantization is not only due to the quantization noise in the weights but also due to the quantization error accumulated in activations as they pass through the layers of networks and that isn't accounted for in this method.

**Questions:**

See weaknesses.

---

### Official Review · Reviewer_wZiN · 2025-10-31

**Soundness:** 2
**Presentation:** 2
**Contribution:** 2
**Rating:** 2
**Confidence:** 4

**Summary:**

This paper proposes a data-free post-training quantization method for Transformers (Data-Free PTQ). By leveraging the parameter space symmetry between the Query and Key weight matrices, an orthogonal transformation is learned to minimize the quantization noise variance. The authors derive a closed-form approximation of the quantization error and optimize it using a rotation matrix, requiring no calibration data. This method can be used as a preprocessing step for existing quantization algorithms and its effectiveness has been verified on BERT and OPT models.

**Strengths:**

1. A data-free quantization method based on parameter symmetry is proposed, which is theoretically innovative and rigorously derived.

2. The paper provides an analytical derivation of the quantization error variance of the QK matrix in Transformer and offers a detailed proof.

3. The method is completely data-independent, lightweight, and only applies an orthogonal transformation to the weight matrix. It requires no additional training or forward computation and is easy to integrate into engineering projects.

**Weaknesses:**

1. The method assumes a standard dot-product attention structure and does not consider rotational positional encodings such as RoPE, therefore it is not applicable to modern mainstream LLMs.

2. The experimental validation scope is narrow, only tested on BERT and OPT-1.3B, lacking validation on newer models such as LLaMA, Qwen, and Mistral.

3. There is a lack of comparison with strong baselines such as GPTQ, AWQ, and SmoothQuant, failing to demonstrate superpositionability or relative advantages.

4. The improvement is limited at low bit depths (e.g., 4-bit) and it is not robust enough for scenarios with high quantization errors.

**Questions:**

1. Can this orthogonal transformation be extended to attention structures that support RoPE or other phase-encoded models?

2. Have you tried applying the same rotation optimization to the Wv and Wo matrices?

3. Can it be applied to models such as LLaMA, Qwen, and Mistral?

4. Are there any additional benefits or conflicts when combined with GPTQ or AWQ?

---

### Official Review · Reviewer_C527 · 2025-11-01

**Soundness:** 2
**Presentation:** 2
**Contribution:** 2
**Rating:** 2
**Confidence:** 4

**Summary:**

This paper proposes a data-free method for post-training quantization of Transformers by leveraging parameter-space symmetry. The core idea is to find an orthogonal transformation of the query and key weight matrices that minimizes an analytically derived estimate of quantization noise, without altering the model's functionality. This method serves as a lightweight, plug-and-play pre-processing step compatible with existing PTQ pipelines. Evaluations on BERT and OPT models for text classification and generation tasks show reduced activation ranges and improved perplexity under quantization.

**Strengths:**

1、The paper presents a novel idea of explicitly minimizing an analytic estimate of quantization noise by optimizing over the orthogonal symmetry of the attention module.

2、The method is designed as a model-agnostic, lightweight pre-processing step, making it easy to integrate into existing quantization pipelines.

3、The method's independence from calibration data is a distinct advantage in data-sensitive scenarios.

**Weaknesses:**

1、The lack of direct comparison with current SOTA PTQ methods makes it impossible to assess the method's relative performance.

2、The experiments do not include larger models (>7B parameters), which are the primary target for efficient quantization, limiting the generalizability of the conclusions.

3、The improvements are negligible on text classification and modest on text generation (especially at low bits), failing to convincingly demonstrate a breakthrough.

4、The empirical evaluation is somewhat narrow. It relies on a limited set of benchmarks and, more critically, does not explore the method's effectiveness under extremely low-bit quantization (e.g., W2A16 or W2A8).

**Questions:**

1、Could you provide results demonstrating that your pre-processing step provides a consistent performance boost when integrated before SOTA PTQ methods like GPTQ? This is essential to validate its utility as a general-purpose module rather than just an improvement over a simple baseline.

2、Why were comparisons with existing SOTA PTQ methods (e.g., GPTQ, SmoothQuant) not included? Could you provide such comparisons in the rebuttal?

3、How does your method perform under more aggressive quantization settings, such as 2-bit for weights (W2A16) or even 2-bit for weights and 8-bit for activations (W2A8)?

4、How does your method scale to models with >10B parameters? Are there any computational or memory bottlenecks during optimization for models with many layers and large hidden dimensions?

5、As shown in the results, the performance gain at 4-bit quantization is relatively small. What do you believe is the root cause? Is it a limitation of the objective function, or is the effect of pre-processing diluted by error propagation in the low-bit regime?

---

### Author Response · Authors · 2025-12-03
**Author Response**

We thank the reviewers for their feedback and insightful comments. We are glad many reviewers found our work theoretically novel. We address specific comments below.

Limited Empirical Evaluation

Several reviewers discuss limited empirical evaluation on both larger models and comparison to current SOTA PTQ methods. These suggestions have been very helpful for evaluating our approach in more practical settings. Due to the flexible, data-free nature of our method, the impact from SOTA PTQ approaches obscured the impacts of our approach. We opted to focus on comparing to standard uniform quantization baselines to observe and isolate the benefits of our pre-processing step.
We are particularly grateful for reviewer suggestions on additional tasks/metrics for evaluating quantization performance. Our empirical evaluation was limited by compute resources but we agree that evaluation on larger models is essential for future experiments.

Applying to RoPE attention

RoPE attention uses a modified attention block with a block rotational matrix $R_m$ used for computing the attention scores. RoPE has different symmetries than SDPA but our general approach can still work by optimizing over the symmetries of RoPE instead of SDPA. The quantization noise estimate is for SDPA so a new estimate would need to be derived for RoPE. Given these two changes, the rest of our algorithm are compatible with RoPE.

---

### Note · Authors · 2025-12-03

I have read and agree with the venue's withdrawal policy on behalf of myself and my co-authors.